# Novel SPEA Superantigen Peptide Agonists and Peptide Agonist-TGFαL3 Conjugate. In Vitro Study of Their Growth-Inhibitory Effects for Targeted Cancer Immunotherapy

**DOI:** 10.3390/ijms241310507

**Published:** 2023-06-22

**Authors:** Sara S. Bashraheel, Sayed K. Goda

**Affiliations:** 1QU Health, Qatar University (QU), Doha P.O. Box 2713, Qatar; sara.bashraheel@qu.edu.qa; 2College of Science and Technology, University of Derby, Derby DE22 1GB, UK

**Keywords:** superantigen-based peptide agonist, superantigen-based peptide conjugate, targeted cancer immunotherapy, cancer combination therapy, SPEA superantigen

## Abstract

Bacterial superantigens (SAgs) are effective T-cell stimulatory molecules that lead to massive cytokine production. Superantigens crosslink between MHC class II molecules on the Antigen Presenting Cells (APC) and TCR on T-cells. This enables them to activate up to 20% of resting T cells, whilst conventional antigen presentation results in the activation of 0.001–0.0001% of the T cell population. These biological properties of superantigens make them attractive for use in immunotherapy. Previous studies have established the effectiveness of superantigens as therapeutic agents. This, however, was achieved with severe side effects due to the high lethality of the native toxins. Our study aims to produce superantigen-based peptides with minimum or no lethality for safer cancer treatment. In previous work, we designed and synthesized twenty overlapping SPEA-based peptides and successfully mapped regions in SPEA superantigen, causing a vasodilatory response. We screened 20 overlapping SPEA-based peptides designed and synthesized to cover the whole SPEA molecule for T-cell activation and tumor-killing ability. In addition, we designed and synthesized tumor-targeted superantigen-based peptides by fusion of TGFαL3 either from the N′ or C′ terminal of selected SPEA-based peptides with an eight-amino acid flexible linker in between. Our study identified parts of SPEA capable of stimulating human T-cells and producing different cytokines. We also demonstrated that the SPEA-based peptide conjugate binds specifically to cancer cells and can kill this cancer. Peptides induce T-cell activation, and tumor killing might pave the way for safer tumor-targeted superantigens (TTS). We proposed the combination of our new superantigen-based peptide conjugates with other immunotherapy techniques for effective and safer cancer treatment.

## 1. Introduction

Cancer is a disease of uncontrolled proliferation of cells that used to be healthy and subject to the controlled mechanism. Traditional cancer treatment strategies have been surgery, chemotherapy, and radiotherapy. Due to the severe side effects and resistance to these treatments, we need more tolerant and effective cancer therapies. The focus has been on utilizing the patient’s immune system to find and destroy cancer cells with little or no harm to the healthy ones. Recently several immunotherapy strategies have been developed, such as immune checkpoint inhibitors, Anti-CTLA-4 antibodies, Anti-PD-1 drugs, and Anti-PD-L-1 drugs [1,2,3,4,5,6,7,8,9,10], chimeric antigen receptor T-cells, or CAR-T [11,12,13,14,15,16,17,18,19,20,21].

Superantigens (SAg) are protein toxins known to activate up to 20% of resting T-cells by binding to MHC-II on Antigen Presenting Cells (APC) and Vβ domain of T-cell Receptor from the side and crosslinking both together [22,23]. This feature makes the SAg very attractive for immunotherapy. Superantigens have been heavily studied for cancer immunotherapy as it is effective in T-cell-dependent tumor killing [24,25,26,27,28,29,30]. To minimize cytotoxicity by the massive immune cell activation in the body, SAgs have been fused to anticancer antibody fragments to specifically target the induced T-cell activity towards cancer cells [31,32,33,34]. This approach is known as tumor-targeted superantigens (TTS). Several studies have linked bacterial SAgs to Fab fragment tumor-specific antibodies or ligands targeting a receptor either highly expressed or only expressed on the tumor cells. Ligand-targeted therapeutics (LTTs) have been favored over mAbs due to their lower antigenicity and higher drug penetration level into a solid tumor [35,36].

Epidermal growth factor receptor (EGFR) is a cancer marker in various carcinomas, including breast, head, neck, oesophagal, gastric, pancreatic, colorectal, prostate, bladder, renal, ovarian, and ovarian Non-Small Cell Lung Cancer (NSCLC). The higher EGFR expression level has been associated with the advanced tumor stage [37,38,39,40]. EGFR-directed antibodies have been used in cancer therapy [41]. Moreover, SAgs were fused with the third loop of transforming growth factor α (TGFαL3) to study the possibility of the therapeutic application of TGFαL3-SAg as a novel anti-tumor candidate in EGFR tumor-expressing cells [37,42,43,44].

One side effect of SAgs is Sepsis Shock Syndrome, which results from cytokine accumulation in the body. In vivo studies demonstrate that the severity of toxic shock is linked to the increased systemic level of cytokines, IL-6 in particular [45,46,47,48]. Therefore, it is reasonable to assume that the reduced cytokine level could lead to reduced toxicity of the SAgs. A point mutation D227A in the MHC-II binding site of SEA resulted in reduced toxicity. Tumor-bearing mice treated with Fab-SEA D227A resulted in a significant tumor reduction and less toxicity compared to Fab-SEA protein. The lower toxicity was due to the low binding affinity to MHC-II, resulting in a lower systemic cytokine level [48,49].

The present study proposes a new approach to produce a superantigen-positive lethality-negative molecule for cancer immunotherapy. We synthesized SPEA superantigen-based peptides and screened for peptide agonists of SPEA. We then show that the isolated peptide(s) could interact with T cell receptors and behave as superantigens but with much less activity and production of cytokines. Therefore, these peptides could enhance the patient’s immune system without causing the cytokine storm, usually caused by the full superantigens.

In this study, the identified peptide agonists are then conjugated with cancer-targeted sequence loop GFαL3. Therefore, the produced novel tumor-targeted SAg-based peptides conjugates could accomplish tolerable cancer immunotherapy.

## 2. Results

### 2.1. T-Cell Activity Assay

SPEA-based peptides were tested for activity by assessing their ability to activate T-cells. PBMCs treated with SPEA-based peptides were stained with fluorescently labelled anti-CD3 and anti-CD25 antibodies and analyzed using a flow Cytometer for CD3+ CD25+ cells. SPEA was used as a positive control, and DMSO and H2O were used as a negative control. Figure 1 shows that peptides SP2, SP3, and SP15 significantly affected T-cells. Up to 30% T-cell activation was achieved in response to SP2, whereas up to 25% T-cell activation was achieved by SP3 and 15% by SP15. All remaining peptides had no significant T-cell activation effect.

### 2.2. Production of Cytokines

In Figure 2A and Figure 3A, the concentration of IL-6 produced in response to SPEA, SP2, SP3, and SP15 is around 31,000, 5900, 2500, and 3000 pg/mL, respectively, whereas the fusion peptides TGFSP2, SP2TGF, TGFSP3, SP3TGF, TGFSP15, and SP15TGF (in all peptide conjugates, TGF = TGFαL3 resulted in 3831 pg/mL, 6365 pg/mL, 5624 pg/mL, 5269 pg/mL, 8387 pg/mL, and 779 pg/mL, respectively. The concentration of IL-10 was reduced from 320 pg/mL in response to treatment with SPEA to around 27 pg/mL in response to SP2 and SP3, and 48 pg/mL in response to SP15 (Figure 2B). The level of IL-10 remained at 48 pg/mL for TGFSP2, but elevated to 65 pg/mL in response to SP2TGF, TGFSP3, and SP3TGF, and 241 pg/mL in response to TGFSP15 (Figure 3B). Furthermore, the concentration of IL-1β was 550 pg/mL in response to SPEA. At the same time, it decreased to around 100 pg/mL in response to SP2 and SP3 (Figure 2D). IL-1β concentration produced in response to TGFSP2, SP2TGF, TGFSP3, SP3TGF, and TGFSP15 was 154 pg/mL, 94 pg/mL, 126 pg/mL, 152 pg/mL, and 282 pg/mL, respectively (Figure 3D). The mixed culture incubated with SP2, SP3, or SP15 did not produce TNFa (Figure 2C). The fusion peptides had a low but significant concentration of TNFa produced, around 87 pg/mL, 45 pg/mL, 41 pg/mL, 24 pg/mL, and 46 pg/mL in response to TGFSP2, SP2TGF, TGFSP3, SP3TGF, and TGFSP15, respectively (Figure 3C). SP15TGF was unable to stimulate significant cytokine production (Figure 3C).

### 2.3. Binding Assessment

The binding of tumor-targeted superantigen-based peptides to EGFR on MDA-MB-468 cells was assessed by ArrayScan™ XTI. SP3TGF and TGFSP15 had the highest percentage of binding to EGFR receptor, up to 45%, whereas SP15TGF had up to 35% binding, and TGFSP2, SP2TGF, and TGFSP3 had 15–25% binding to EGFR receptors (Figure 4).

### 2.4. MTT Assay

Compared to the Media control, the SPEA-treated PBMCs showed a higher absorbance than the control due to a proliferation effect of the whole superantigen, while SP2TGF and SP3TGF had no toxic effect on the PBMC viability. On the other hand, TGFSP2 and TGFSP3 had decreased cell viability compared to the media (Figure 5).

### 2.5. Tumor Killing Assay

Tumor cell viability was measured by counting DAPI positive nuclei for each sample as a percentage of TP (Tumor cells with PBMCs) sample. Around 50% tumor killing was detected in response to SP2 and SP3 compared to the corresponding solvent control. At the same time, SP15 had no effect, and SPEA showed 70% tumor killing (Figure 6). Tumor cell viability was also assessed in response to tumor-targeted peptides. TGFSP2 shows the highest percentage of tumor cell count reduction with up to 65% tumor killing. Further, 20–40% killing was achieved in response to SP2TGF, TGFSP3, and SP3TGF. Interestingly, TGFSP15 induced a 25% tumor-killing effect (Figure 7).

## 3. Discussion

The cytotoxicity and pathogenicity of bacterial superantigens, including SPEA, are due to their ability to activate up to 20% of the T-cell repertoire. This strong interaction is the main reason for its mitogenic activity. These biological properties of superantigens make them very attractive for use in immunotherapy. Previous studies have shown the effectiveness of different superantigens as therapeutics for cancer immunotherapy [27,32,50,51,52,53]. These studies used the entire superantigen molecules, leading to many treatment pitfalls. The complete molecules raised high immunogenicity, followed by neutralization by the patient’s immune system, demolishing its therapeutic effect [52]. The large size of the full superantigen molecules would also be an obstacle to the permeation of the biological barrier in the patient’s body. This is in addition to the severe toxicity which hindered their use in clinical trials. We embarked on a campaign to produce modified and safer forms of superantigens and to test their ability to enhance the immune systems for cancer therapy.

One of the damages bacterial superantigens can cause is severe hypotension, possibly leading to the patient’s death. Our previous work [54] identified the regions of SPEA involved in causing vasodilation and possible hypotension. We carried out the work by designing and synthesizing a series of 20 overlapping peptides spanning the entire sequence of SPEA. The vascular response of each peptide was measured, and three active peptides (SP7, SP11, and SP19) were identified [54]. Removing these sequences and the production of modified superantigens lacking these peptides could lead to safer SPEA variants for cancer treatment.

To produce superantigenicity-positive lethality-negative molecules, we extended our work further. We tested the following 20 overlapping peptides, which were synthesized in our previous work [54], which cover the full amino acid sequence of SPEA for T-cell activation and production of cytokines:

SP1(1–40), SP2(11–50), SP3(21–60), SP4(31–70), SP5(41–80), SP6(51–90), SP7(61–100), SP8(71–110), SP9(81–120), SP10(91–130), SP11(101–140), SP12(111–150), SP13(121–160), SP14(131–170), SP15(141–180), SP16(151–190), SP17(161–200), SP18(171–210), SP19(181–220), and SP20(191–222).

We successfully identified the fragments SP2(11–50), SP3(21–60), and SP15(141–180), which can stimulate the proliferation of human T-cells and produce the tumor necrosis IL-1, IL-6, and IL-10 (Figure 1), and show similar profile as noted by the full SPEA superantigen. The level of cytokines, however, produced is lower than the full superantigen. It was reported that the amino acid sequence TNKKMVTAQELD is the sequence on the superantigen that binds to CD28, and is essential for the induction of cytokines. This sequence is not entirely included in the SP2 and SP3 peptides, which might be the reason for their reduced superantigenicity and toxicity [55]. Therefore, the peptides isolated in this study would avoid the super toxicity caused by the complete superantigen molecules, but still enhance the T-cell proliferation for cancer immune therapy.

IL-6 is a cytokine with pleiotropic activities. It plays a vital role in acquiring immune response by stimulating antibody production and effective T-cell development. IL-6 Figures tumor growth by mobilizing anti-tumor T-cell immune response to control tumor growth. IL-6 plays a crucial role in boosting T-cell trafficking to lymph nodes and tumor sites. IL-6 signaling can also reshape the T cell immune response, shifting it from a suppressive to a responsive state that can effectively act against tumors [56,57]. On the other hand, because of its pleiotropic nature, continuous dysregulated production can develop different diseases [56].

Studies have shown that IL-1α inhibits cell proliferation of MCF-7 breast cancer, A375 melanoma, prostate stem cells, and murine primary mammary cells by causing G0–G1 cell cycle arrest [58,59,60]. Another investigation showed that IL-1 increases the recruitment of neutrophils to the tumor, which increases the neutrophil ability of the tumor killing. The study suggests that due to its action on neutrophils, IL-1 has a robust anti-cancer effect [61]. IL-10 is an anti-inflammatory cytokine that plays a crucial role in tumor killing and inhibits the production of IFNg, IL-2, IL-3, and TNFα [62].

Our findings so far show that three of the peptides SPEA agonists have similar profiles of the full superantigen SPEA in activating the T- cells and producing different types of cytokines with an anti-cancer effect such as IL-1, IL-6, and IL-10, as mentioned above (Figure 2 and Figure 3). These peptides, however, might not have the severe toxicity of the full superantigens. The significant advantage of the peptide’s agonists identified in this study, it would be much easier to modify these peptides to obtain shorter or mutated peptides to produce more active and efficient variants with less toxicity to enhance T-cells against cancer.

There are two main strategies for drug delivery in cancer treatment, aiming to maximize the drug to target the tumor and minimize the exposure of the healthy cells. One uses a delivery vehicle such as nanoparticles [63,64]. The second is a covalent chemical binding of the leading drug to a small moiety which binds specifically to cancer receptors, such as antibody-drug conjugates (ADC) [65], antibody-directed enzyme prodrug therapy (ADEPT) [66], and many others [67]. We, in this study, implemented the second strategy to maximize the superantigen peptides in the cancer tissue.

Using the second strategy, we extended our work to target tumors using these peptides. Epidermal growth factor receptor (EGF), the human EGF receptor, plays an essential role during cell growth [68]. Studies reported that EGFR is overexpressed in cancer cells such as colorectal, breast, and non–small cell lung cancers [69,70,71]. Due to its overexpression on cancer cells, several investigations used the third loop (L3) of transforming growth factor-α (TGF-α) to conjugate with protein antibodies [72,73,74,75], or full superantigen SEB to exert the toxic effect of a drug in the vicinity of cancers to achieve targeted cancer therapy [76].

To accomplish specificity and minimize toxicity, the SPEA peptide agonists were conjugated by the highly expressed TGFαL3. This targeted moiety will lead the peptide agonist to bind specifically to the cancerous cells, activate a safe number of T lymphocytes, and direct them only to the tumor. This will achieve the specific killing of the tumor and leave the healthy cells intact. Considering the misfolding the loop may cause to the peptide agonist, we designed and synthesized TGFaL3-Peptide conjugate (N-terminal) and Peptide-TGFaL3 conjugate (C-terminal) (Figure 8). These conjugates comprise the third loop of transforming growth factor-α (TGF-α), a flexible linker and the peptide agonist (Figure 8 and Table 1). The results in Figure 6 and Figure 7 show that TGFαL3-peptides agonists and free peptide agonist induced cancer-killing, while the results in Figure 6 show that SP2TGF and SP3TGF had no toxic effect on the healthy PBMC viability. Peptide conjugates; TGFSP2 and TGFSP3 however, had decreased cell viability compared to the media.

This work paves the way that the TGFaL3-Peptide conjugates produced in this work could be used for targeting the treatment of EGFR-expressing cancer cells (Figure 9), and will build the ground for in vivo investigations of these novel conjugates.

Developing combined therapies or strategies for treating oncological diseases is a promising approach to improving the effectiveness of antitumor treatment [77].

The combination may produce an additive or synergistic effect and reduce each component’s therapeutic dose.

Several studies reported that combination therapies for oncological diseases had been shown that it is more effective and achieved higher overall survival than individual treatment alone in different types of cancer, including breast cancer [78] and lung cancer [79]. Previous investigations have demonstrated robust tumor immunity against murine melanoma obtained by combining DNA vaccination with dual CTLA-4 and PD-1 blockade [80]. Combining the traditional cancer treatment of radiotherapy with immunotherapy can produce synergistic effects because of the specific interaction between the immune system and radiation [81,82]. A recent in vitro study has shown that a combination of telomerase inhibition and NK cell therapy increased breast cancer cell line apoptosis [83].

Our novel superantigen agonists peptides conjugate add one more robust strategy to be used in combination with other recent cancer immune therapy strategies. We propose that our new superantigen-peptide conjugates could be combined with other immunotherapies for effective cancer treatment (Figure 10).

## 4. Method and Materials

### 4.1. Reagents, Antibodies, and Kits

SPEA SAg gene with accession number (KY594414) was synthesised by Geneart GmbH (Regensburg, Germany) and previously cloned and expressed in *E. coli*. SPEA-based peptides were synthesized and supplied by GenScript (Piscataway, NJ, USA). MDA-MB-468 cell line was obtained from ATCC collections, MDA-MB-468 (ATCC^®^ HTB-132™). Fetal bovine serum, L-Glutamine 200 mM (100×), and Dulbecco modified eagle’s medium (DMEM) were purchased from Life Technologies Co., Paisley, UK. Penicillin-streptomycin and Dulbecco’s phosphate buffered saline (DPBS) were purchased from Sigma-Aldrich Chemie GmbH, Taufkirchen, Germany; DAPI (40′,6-diamidino-2-phenylindole) for nucleic acid staining was purchased from Thermo-Fisher Scientific, Waltham, MA, USA. Lonza Trypsin/EDTA (10×) was purchased from SLS life sciences Co., Warwick, UK. Custom ProcartaPlex Multiplex Panel targeting IL-1β, IL-2, IL-6, IFNγ, TNFα, and IL-10 was purchased from Thermo-Fisher Scientific. Human anti-CD3-APC IgG1 and anti-CD25-FITC were purchased from ImmunoTools (Friesoythe, Germany). Anti-EGFR antibody [EP38Y] and Goat Anti-Rabbit IgG H&L (FITC) were from Abcam (Cambridge, UK).

### 4.2. T-Cell Activation Assay

Peripheral Blood Mononuclear Cells (PBMC) were freshly isolated from human blood using Lymphoprep. Then, 2.5 × 10^5^ cells/well were seeded into a 96-well plate and were treated with 200 ug/mL of peptides, 800 ug/mL endotoxin-free SPEA, or the solvent controls. Cells were harvested after 72 h incubation at 37 °C, 5% CO_2_, and stained with fluorescently labelled anti-CD3 and anti-CD25 antibodies. T-cell activation analysis was performed using an Accuri C6 flow cytometer (BD Biosciences) by gating for the lymphocyte subset (Appendix A) and measuring the percentage of CD3+ CD25+ cells. The ability of the labelled peptides to activate T-cells was investigated.

### 4.3. Design of SPEA-Based Tumor-Targeted SAg Peptide

To design tumor-targeted superantigen-based peptides, we fused selected peptides with TGFαL3 either from the N′ or C′ terminal of the SPEA-based peptides (Figure 8). Eight-amino acid flexible linker (GGSGSGGG) was introduced between the peptides and the TGFαL3 (VCHSGYVGARCEHADLL). The designed peptides (Table 1) were chemically synthesized by ProteoGenix (Schiltigheim, France). The peptides were dissolved in Acetonitrile as recommended by ProteoGenix.

### 4.4. Production of Cytokines

The cell culture supernatant from above was used to measure the level of different cytokine production in response to SPEA-based peptides and peptide-conjugates, SPEA, or controls by Elisa Assay. Custom ProcartaPlex Multiplex Panel Elisa kit targeting IL-1β, IL-2, IL-6, IFNγ, TNFα, and IL-10 was used as instructed in the manual provided with the kit. The Elisa plate was measured using FlexMap 3D instrument (Luminex, Austin, TX, USA).

### 4.5. Tumor Cell Binding Assay

The binding of tumor-targeted superantigen-based peptides to EGFR on MDA-MB-468 cells was assessed by measuring its ability to block anti-EGFR antibodies from binding to the EGF receptor. To achieve that, 1 × 10^4^ cells were seeded into 96-well plates and incubated overnight at 37 °C, 5% CO_2_. After removing the media, the cells were fixed with 3.8% formaldehyde solution in PBS for 15 min at RT. The mixture was removed and the cells were washed with PBST and blocked with 3% BSA in PBS for 30 min at RT. The cells were washed again with PBST and incubated with 200 ug/mL of fusion peptides or media control for 2 h at RT. After three washes with PBST, cells were incubated with an Anti-EGFR antibody (Abcam) for 1 h at RT and then washed three times with PBST. Finally, cells were stained with Goat Anti-Rabbit IgG H&L-FITC and DAPI to stain the cell nuclei for 30 min. ArrayScan™ XTI (Thermo Fisher Scientific, Waltham, MA, USA) was used for automated quantitation of DAPI and FITC positives. The binding of peptides to tumor cells was measured by:

% Binding of peptide to tumor cells = (1 − (response of tumor cell treated with superantigen-based tumor-targeted peptides/response of tumor cell treated with anti-EGFR)) × 100

### 4.6. MTT Assay

The Fusion peptides were evaluated for their toxicity on healthy cells using an MTT assay. PBMCs were isolated and seeded into a 96-well plate (2.5 × 10^5^ cells/well). Cells were incubated with fusion peptides, endotoxin-free SPEA, or the solvent controls for 24 h at 37 °C and under 5% CO_2_. Cells were then treated with 20 uL of 5 mg/mL MTT solution (Sigma-Aldrich, St. Louis, MO, USA) and incubated for 4 h at 37 °C and under 5% CO_2_. The precipitate was dissolved with 200 uL of 20% SDS, and the absorbance was measured at 570 nm. The media (untreated cells) sample is considered 100% viable.

### 4.7. Assessment of Tumor Killing Ability

Human breast carcinoma cell MDA-MB-468 (obtained from the ATCC: ATCC^®^ HTB-132™) were seeded in a 96-well plate (6.4 × 10^3^ cell/well) and incubated at 37 °C in 5% CO_2_ overnight to allow cell attachment to the well. Cells were treated with 800 ug/mL of endotoxin-free SPEA superantigen, 200 ug/mL SPEA-based peptides, or tumor-targeted superantigen-based peptides. Then, 3.2 × 10^4^ PBMCs were added to wells with tumor cells and SAg derivatives, and incubated for 48 h at 37 °C incubator with 5% CO_2_. The cell lysate was removed and used for cytokine assessment. Cells were fixed with 3.8% Formaldehyde solution for 10 min and then stained with DAPI. The number of viable tumor cells was measured by ArrayScan™ XTI (Thermo Fisher Scientific, Waltham, MA, USA) where an automated quantitation of DAPI positive nuclei was performed using the target activation module, and the apoptotic nuclei were distinguished and excluded by their collapsed size, higher chromatin intensities, and nuclear fragmentation [84]. The PBMCs were also excluded by size. Cell count is presented as a percentage of tumor + PBMCs (TP) will count.

### 4.8. Statistical Analyses

Data are presented as mean ± standard error of the mean (S.E.M) of “n” observations. All graphs were constructed using GraphPad Prism 5 software (San Diego, CA, USA). Statistical analysis was performed using Student’s *t*-test as appropriate. *p* values < 0.05 were considered statistically significant.

## 5. Conclusions

Our work produced novel superantigen SPEA-peptide agonists and SPEA-peptide agonist-EGF conjugates for the targeted treatment of EGFR-expressing cancers. We showed that these peptides and their conjugates produced different cytokines and effectively killed cancer cells, while peptides and peptide-conjugates have no effect on the healthy PBMCs. The great advantage of our work is the ability to carry out mutation studies on our isolated peptide agonists to produce more effective superantigen peptides. Further, to fuse our peptides with other cancer-specific moieties to target treating other cancers. This study would pave the way for in vivo investigation on animals and humans.

## Figures and Tables

**Figure 1 ijms-24-10507-f001:**
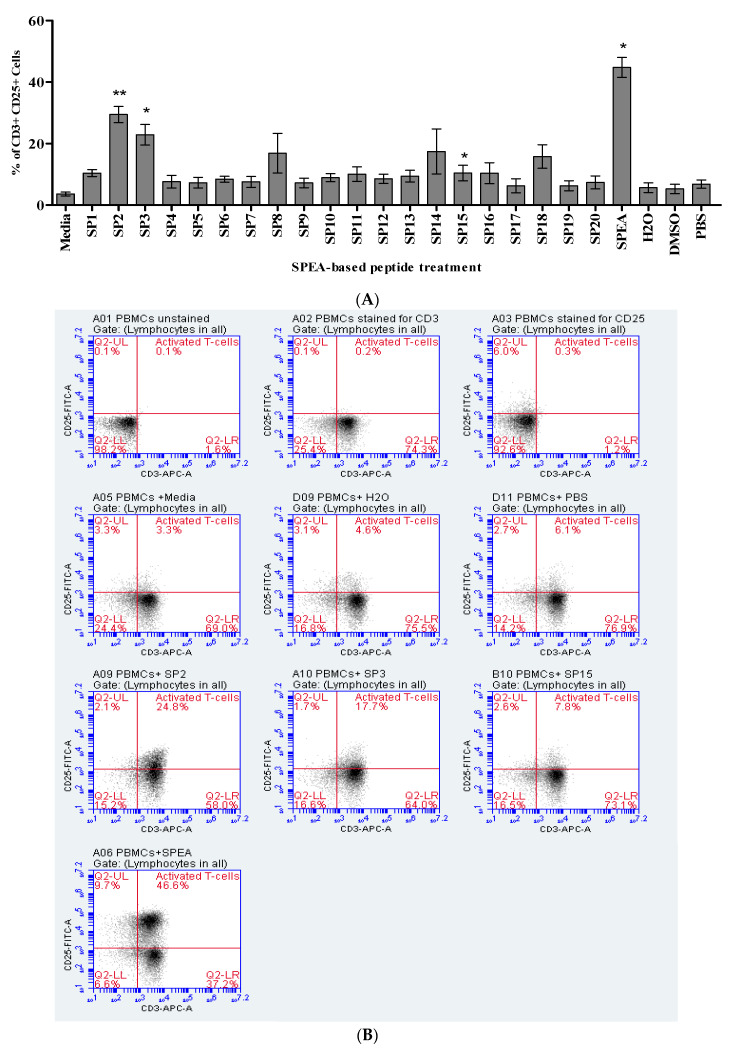
Percentage of CD3+ CD25+ Cells (activated T-cells) in response to treatment with SPEA-agonist peptides. (**A**) Data of 20 SPEA-based peptides presented as mean percentage of CD3+ CD25+ cells ± standard error of the mean (S.E.M) of 3 independent experiments (*n* = 3), * *p* ˂ 0.05, ** *p* ˂ 0.01, compared to the corresponding solvent control. (**B**) Representative flow cytometric dot-plots CD3/CD25 staining in which cells were pre-gated for lymphocytes.

**Figure 2 ijms-24-10507-f002:**
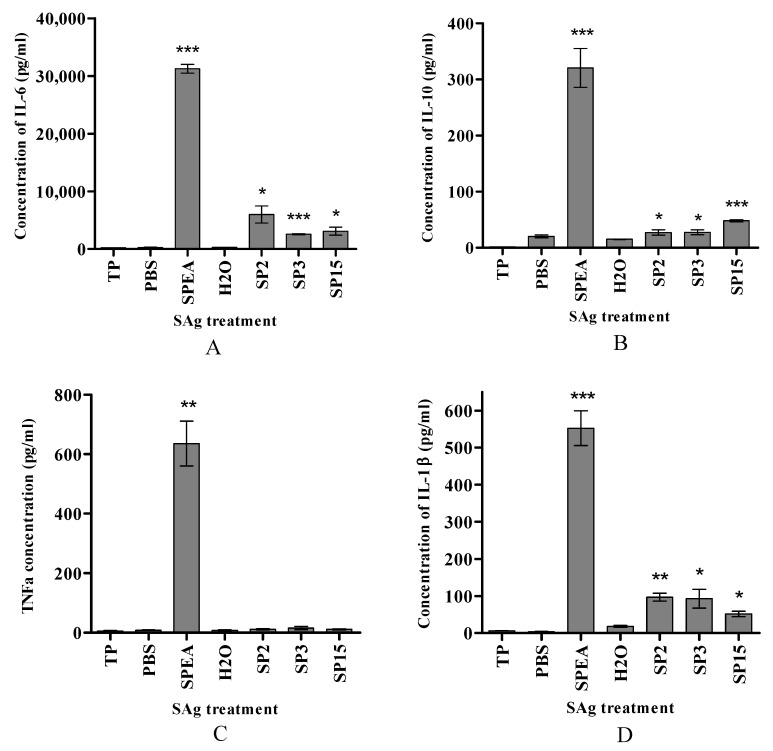
Cytokine production in response to SPEA-based peptides in a mixed culture of PBMCs and MDA-MB-468 cells. (**A**) concentration of IL-6, (**B**) concentration of IL-10, (**C**) concentration of TNFα, and (**D**) concentration of IL-1β. Data presented as a mean concentration in pg/mL ± standard error of the mean (S.E.M) of 3 experiments (*n* = 3).* *p* ˂ 0.05, ** *p* ˂ 0.01, *** *p* ˂ 0.001, compared to treatment with the corresponding solvent (H_2_O for SP2, SP3, and PBS for SPEA).

**Figure 3 ijms-24-10507-f003:**
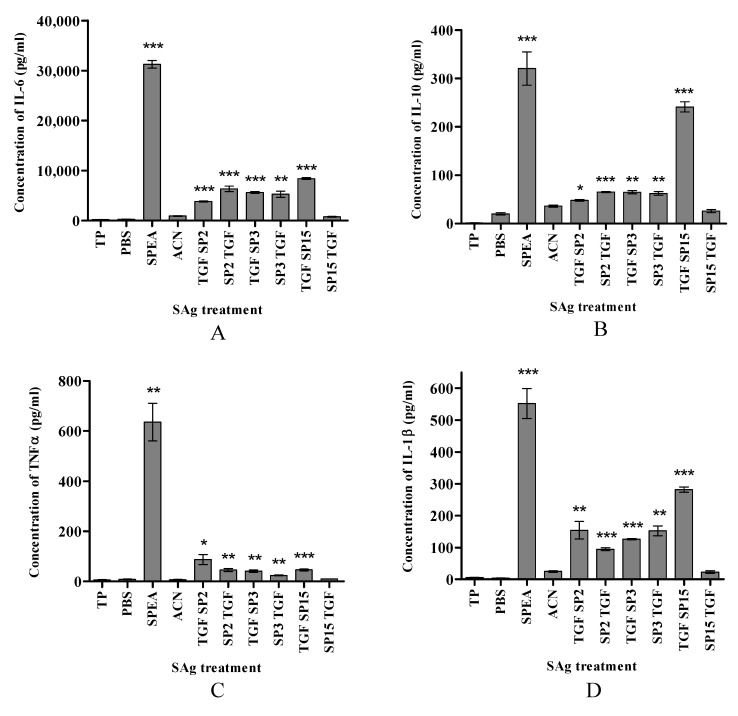
Cytokine production in response to SPEA-based tumor-targeted peptides in a mixed culture of PBMCs and MDA-MB-468 cells. (**A**) concentration of IL-6, (**B**) concentration of IL-10, (**C**) concentration of TNFα and (**D**) concentration of IL-1β. Data presented as a mean concentration in pg/mL ± standard error of the mean (S.E.M) of 3 experiments (*n* = 3). * *p* ˂ 0.05, ** *p* ˂ 0.01, *** *p* ˂ 0.001, compared to treatment with the corresponding solvent (ACN for Fusion peptides and PBS for SPEA).

**Figure 4 ijms-24-10507-f004:**
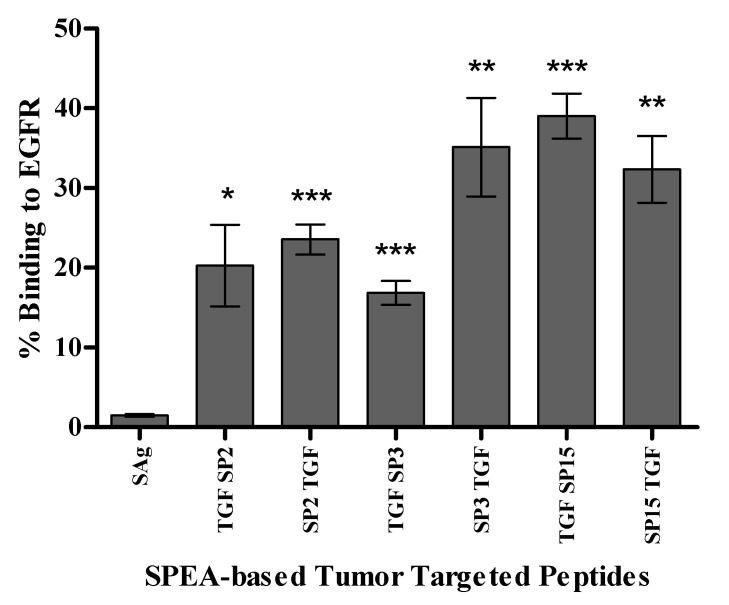
SPEA-based tumor-targeted peptides are binding to EGFR. Data of SPEA-based peptides presented as mean percentage of tumor cells treated with anti-EGFR antibody ± standard error of the mean (S.E.M) of 3 experiments (*n* = 3), * *p* ˂ 0.05, ** *p* ˂ 0.01, *** *p* ˂ 0.001, compared to treatment with full superantigen.

**Figure 5 ijms-24-10507-f005:**
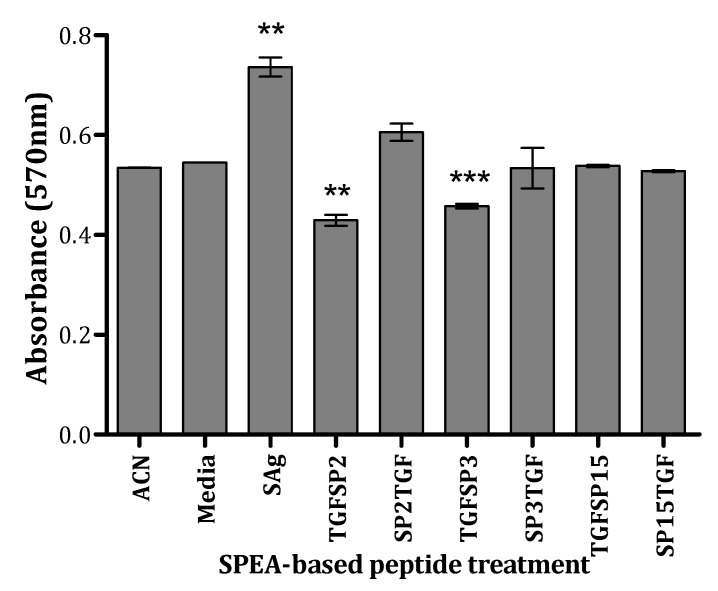
Effect of SPEA-based tumor-targeted peptides on PBMCs viability. Data presented as mean absorbance ± standard error of the mean (S.E.M) of 3 experiments (*n* = 3). ** *p* ˂ 0.01, *** *p* ˂ 0.001, compared to media control).

**Figure 6 ijms-24-10507-f006:**
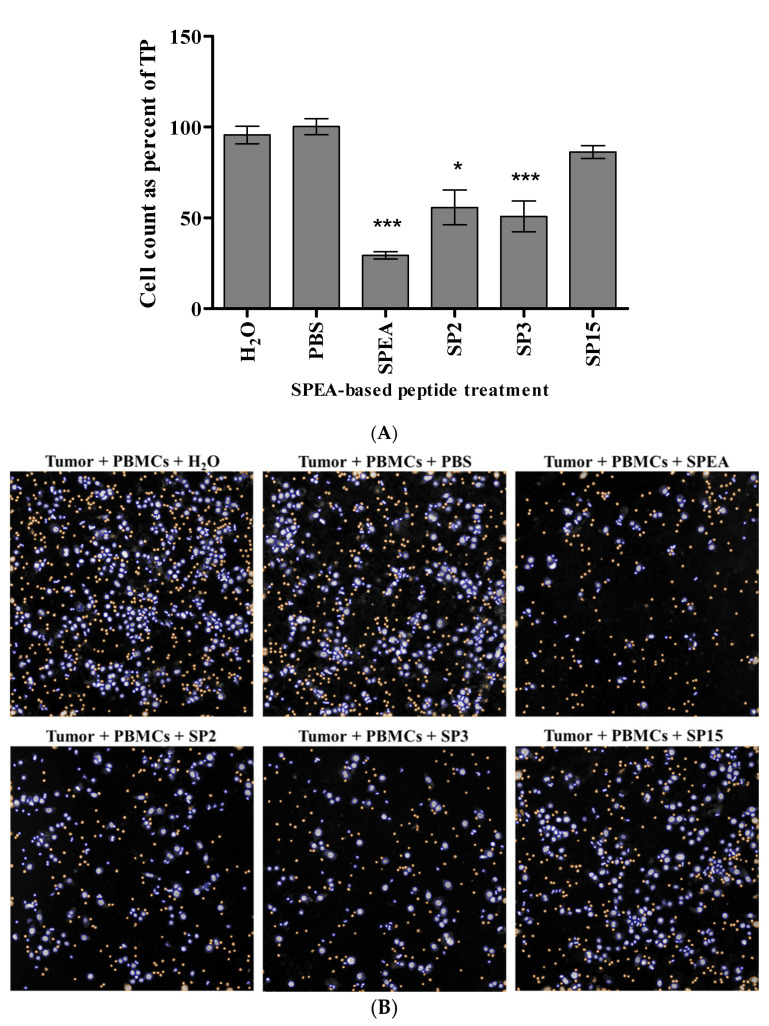
Effect of different SPEA-based peptides on MDA-MB-468 cell viability in a mixed culture with PBMCs. (**A**) Data presented as the mean of DAPI positive nuclei as a percentage of TP ± standard error of the mean (S.E.M) of 6 independent experiments (*n* = 6). * *p* ˂ 0.05, *** *p* ˂ 0.001, compared to treatment with corresponding solvent. (**B**) Representative Images of the DAPI-stained nuclei of viable cells after incubation with the different peptides. The blue color is DAPI positive MDA-MB-468 cells, whereas the orange color is for the excluded PBMCs.

**Figure 7 ijms-24-10507-f007:**
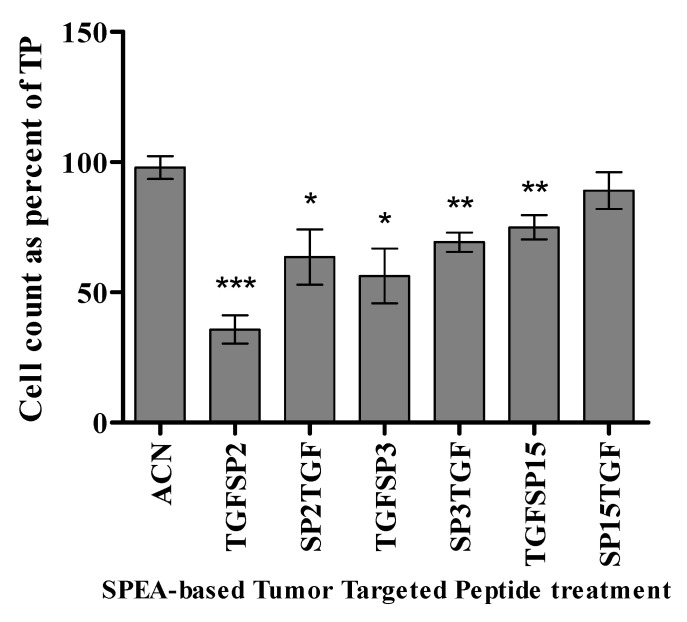
Effect of different SPEA-based tumor-targeted peptides on MDA-MB-468 cell viability in a mixed culture with PBMCs. Data presented as the mean of DAPI positive nuclei as a percentage of TP ± standard error of the mean (S.E.M) of 6 independent experiments (*n* = 6). * *p* ˂ 0.05, ** *p* ˂ 0.01, *** *p* ˂ 0.001, compared to treatment with the solvent (ACN).

**Figure 8 ijms-24-10507-f008:**
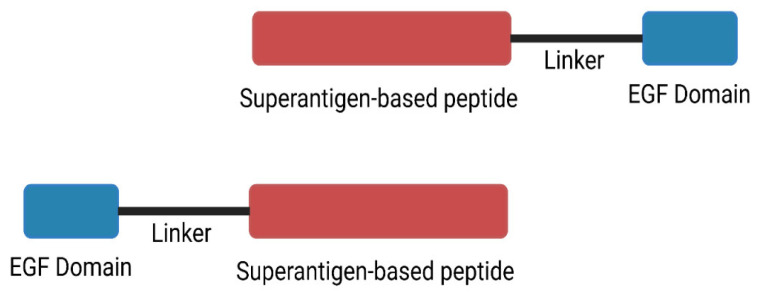
Synthesis of superantigen agonist peptide EGF conjugated at either N-terminal or C-terminal.

**Figure 9 ijms-24-10507-f009:**
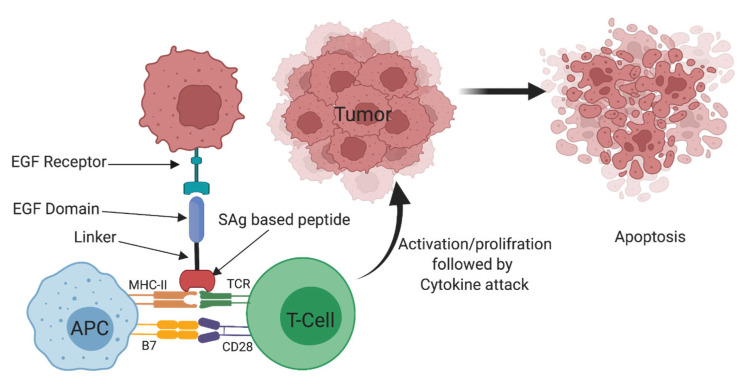
Targeted cancer treatment using superantigen agonist conjugated peptide, modified [77].

**Figure 10 ijms-24-10507-f010:**
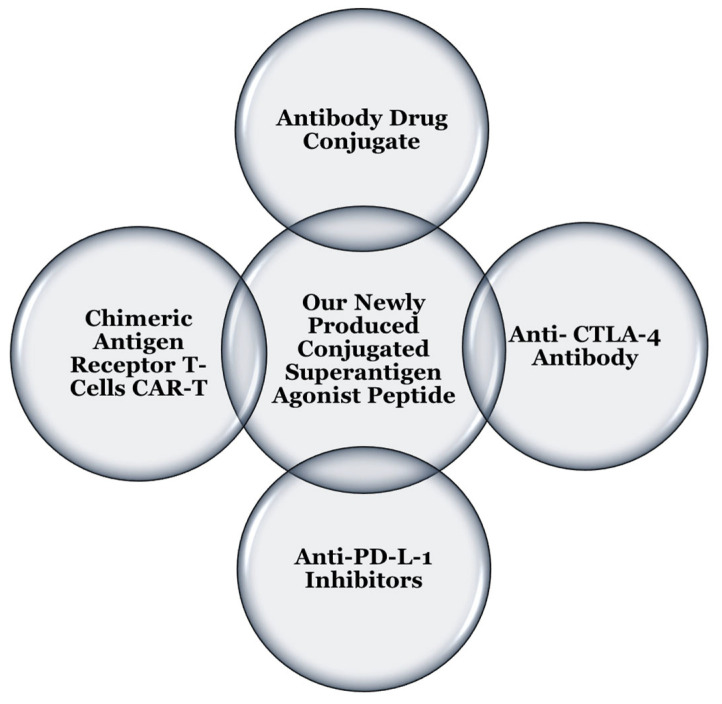
Possible combinations of other immunotherapy techniques with our newly superantigen agonist peptide conjugates to overcome drug resistance.

**Table 1 ijms-24-10507-t001:** SPEA-based tumor-targeted SAg peptide Sequence.

SP2TGF	HRSSLVKNLQNIYFLYEGDPVTHENVKSVDQLLSHDLIYNGGSGSGGGVCHSGYVGARCEHADLL
TGFSP2	VCHSGYVGARCEHADLLGGSGSGGGHRSSLVKNLQNIYFLYEGDPVTHENVKSVDQLLSHDLIYN
SP3TGF	NIYFLYEGDPVTHENVKSVDQLLSHDLIYNVSGPNYDKLKGGSGSGGGVCHSGYVGARCEHADLL
TGFSP3	VCHSGYVGARCEHADLLGGSGSGGGNIYFLYEGDPVTHENVKSVDQLLSHDLIYNVSGPNYDKLK
SP15TGF	VTAQELDYKVRKYLTDNKQLYTNGPSKYETGYIKFIPKNKGGSGSGGGVCHSGYVGARCEHADLL
TGFSP15	VCHSGYVGARCEHADLLGGSGSGGGVTAQELDYKVRKYLTDNKQLYTNGPSKYETGYIKFIPKNK

## Data Availability

All data are available with the authors and will be provided if requested.

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
