# Peer review of "Novel SPEA Superantigen Peptide Agonists and Peptide Agonist-TGFαL3 Conjugate. In Vitro Study of Their Growth-Inhibitory Effects for Targeted Cancer Immunotherapy"

_ijms, 2023, doi:10.3390/ijms241310507_

Round 1
Reviewer 1 Report
The authors in the article titled: “Novel SPEA Superantigen Peptide Agonists and Peptide Agonist-TGFαL3 Conjugate. In vitro study of their growth-inhibitory effects for targeted cancer immunotherapy.” presented the continuous of their previous investigation. The article can be published in IJMS in its present form, after minor typos corrections:
Line 128 instead of “37C” it should be 37°C
Line 317 instead of “Il-10” it should be IL-10
Line 345 instead of “liker” should be linker
*Additional comments from reviewer
The authors in their paper titled: Novel SPEA Superantigen Peptide Agonists and Peptide Agonist-TGFαL3 Conjugate. In vitro study of their growth-inhibitory effects for targeted cancer immunotherapy gave interesting approach to superantigen SPEA-peptide agonist and SPEA-peptide agonist-EGF conjugates, which can be very useful in future cancer treatments. This topic is definitely interesting in this field. The presentation of the results can be better organized, especially giving attention to the names of superantigen peptide agonists and conjugates.
I have no complaints about the quality of English language.
Author Response
Reviewer 1
We thank you for your time and attention in reviewing our paper and sending your report.
This is our response to the points made:
The authors in the article titled: “Novel SPEA Superantigen Peptide Agonists and Peptide Agonist-TGFαL3 Conjugate. In vitro study of their growth-inhibitory effects for targeted cancer immunotherapy.” presented the continuous of their previous investigation. The article can be published in IJMS in its present form, after minor typos corrections:
Thank you.
Line 128; instead of “37C” it should be 37°C
Is done, line 365 in the modified manuscript; thank you
Line 317; instead of “Il-10” it should be IL-10
Is done; line 255 in the modified manuscript; thank you
Line 345; instead of “liker” should be linker
Is done, line 281 in the modified manuscript; thank you
*Additional comments from the reviewer
The authors in their paper titled: Novel SPEA Superantigen Peptide Agonists and Peptide Agonist-TGFαL3 Conjugate. In vitro study of their growth-inhibitory effects for targeted cancer immunotherapy gave interesting approach to superantigen SPEA-peptide agonist and SPEA-peptide agonist-EGF conjugates, which can be very useful in future cancer treatments. This topic is definitely interesting in this field. The presentation of the results can be better organized, especially giving attention to the names of superantigen peptide agonists and conjugates.
We made some corrections to the results and added a notion to clarify the peptide conjugates names. We thank you
Reviewer 2 Report
This manuscript covers an interesting topic, however there are a few major issues, which should be addressed first.
Table 1 will be better suited for supplementary materials.
The super antigen, please comment in which diseases this super antigen is expressed, thus the relevance to clinics will be pointed.
Your flow gating is a bit of confusing. did you check for liver/dead. dis you check for single cells. please present in supplementary your full gating strategy. was the voltage similar across all samples. how many cells were originally in each well ?
figure 7 - what methods was used ? which microscope w=for this scan ?
Il-6 and IL-1 are well known pro-inflammatory markers. are there any other cytokines which are related to your findings ?
English language is fine, a few moderate spelling issues were spotted.
Author Response
Reviewer 2
We thank you for your time and attention in reviewing our paper and sending your report.
This is our response to the points made:
This manuscript covers an interesting topic, however there are a few major issues, which should be addressed first.
Thank you for your kind comment
Table 1 will be better suited for supplementary materials.
Thank you, it might be better to keep Table 1 and fig. 8 together for the reader. We, however, are happy to move it to supplementary materials.
The super antigen, please comment in which diseases this super antigen is expressed, thus the relevance to clinics will be pointed.
Superantigens (SAgs) are a class of antigens that cause non-specific activation of T-cells resulting in polyclonal T-cell activation and massive cytokine release and causing symptoms similar to sepsis.
Your flow gating is a bit of confusing. did you check for liver/dead. dis you check for single cells. please present in supplementary your full gating strategy. was the voltage similar across all samples. how many cells were originally in each well ?
We did not check for live or dead cells in this experiment; however, we carried MTT assay on the fusion peptides to check their cytotoxicity (fig5). Figure 2B was modified with changes in the labels to make it easier for the reader. Supplementary Figure S1 is added for the gating of lymphocytes as requested. We originally seeded 2.5x105 cells/well; however, the flow cytometer was set to count 10,000 events ungated in each sample. The settings of the number of cells and the voltage were the same for all samples; all the above was mentioned in the methods and materials section. We added Figure S1 to the supplementary for gating.
figure 7 - what methods was used ? which microscope w=for this scan ?
As mentioned in the method section, we used ArrayScan™ XTI (Thermo Fisher Scientific, Waltham, MA, USA) to acquire the cell images and the cell counts. We used the same method by Ismail et al. for cell count (reference 85 in the manuscript).
Il-6 and IL-1 are well known pro-inflammatory markers. are there any other cytokines which are related to your findings ?
Yes, other cytokines are produced, as shown in the manuscript IL-10, TNFα and IL-1β are also produced by our peptide conjugates.
Round 2
Reviewer 2 Report
Authors have addressed previously raised comments.
Moderate english language check is required